# PRIME (Positive Transitions Through the Menopause) Study: a protocol for a mixed-methods study investigating the impact of the menopause on the health and well-being of women living with HIV in England

Shema Tariq,[1] Fiona M Burns,[1,2] Richard Gilson,[1] Caroline Sabin[1]

[1]Institute for Global Health, University College London, London, UK
[2]Royal Free London NHS Foundation Trust, London, UK

**Correspondence to**
Dr Shema Tariq;
s.tariq@ucl.ac.uk

## ABSTRACT

**Introduction** Advances in antiretroviral therapy have transformed HIV into a long-term condition with near-normal life expectancy for those in whom viral replication is well controlled on treatment. This means that age-related events, including menopause, is of increasing importance in the care of people living with HIV. The PRIME (Positive Transitions Through the Menopause) Study aims to explore the impact of the menopause on the health and well-being of women living with HIV (WLHIV).

**Methods and analysis** The PRIME Study is a multicentre, mixed-methods observational study deploying a multiphase sequential design with explanatory and exploratory phases. Phase 1 comprised three focus group discussions with WLHIV. In phase 2 we aimed to administer questionnaires comprising detailed assessment of menopausal status and symptoms to 1500 WLHIV aged 45–60 attending HIV clinics in England. Phase 3 comprised semistructured interviews with a subsample of phase 2 participants. Ongoing quantitative follow-up of 100 participants is planned between October 2018 and September 2019. Qualitative and quantitative data will be kept analytically distinct and analysed using appropriate methods. We will integrate quantitative and qualitative findings using coding matrices.

**Ethics and dissemination** The PRIME Study has ethical approval from the South East Coast-Surrey Research Ethics Committee on behalf of all National Health Service (NHS) sites, and approval from University College London Research Ethics Committee for qualitative work conducted in non-NHS sites. In conjunction with the study Expert Advisory Group (which includes WLHIV), we have drafted a dissemination strategy that takes into account a wide range of stakeholders, including patients, policy makers and healthcare providers. This includes at least five empirical research papers to be submitted to peer-reviewed journals, as well as an accessible report aimed primarily at a non-technical audience (published in May 2018 and launched at a live-streamed event). Both quantitative and qualitative data are held by the PRIME Study team and are available by request.

## Strengths and limitations of this study

► A key strength of the PRIME (Positive Transitions Through the Menopause) Study lies in its combination of quantitative and qualitative data on reproductive ageing in women living with HIV.

► Linking to clinical records (with consent) provides further robust data on clinical outcomes.

► Patient and public involvement is core to the PRIME Study, with women living with HIV involved in study design, recruitment, data collection, data analyses and dissemination.

► In recruiting solely from National Health Service sites in phases 2 and 3, we will have excluded women who are not engaged in HIV care, whose needs and experiences may differ.

► The cross-sectional nature of the baseline PRIME Study means we are unable to infer causality in the analyses of these data; however, longitudinal follow-up is ongoing for a subgroup of participants.

## INTRODUCTION

Advances in antiretroviral therapy (ART) have transformed HIV into a long-term condition with near-normal life expectancy for those in whom viral replication is well controlled on treatment.[1] Consequently, this is leading to a shift in the age distribution of people living with HIV (PLHIV), with nearly two-fifths of people accessing HIV care in the UK aged 50 or over.[2]

Approximately 10 500 women of potentially menopausal age (45–56 years) attended for HIV care in the UK in 2016, a fivefold increase over a 10-year period (Z Yin, Public Health England, personal communication, 3 October 2017). Based on the age distribution of women attending for HIV care in the UK in 2013, a further 10 000

are likely to reach menopausal age by 2023.[3] Globally over half of the 36.7 million PLHIV internationally are female,[4] and the proportion of those aged 50 years and older is increasing.[5] Age-related events (including the menopause) are therefore of increasing importance in the clinical care of women living with HIV (WLHIV).

Natural menopause occurs at a median age of 51 years in the UK,[6] with two-thirds of women reporting symptoms such as hot flushes, sleep disturbance and mood changes,[7] lasting a median duration of 7.4 years.[8] These symptoms are known to impact women's quality of life (QoL), work performance and social lives; over half of the respondents in a recent British survey reported that the menopause had negatively impacted their lives.[9]

Women living with long-term conditions, such as HIV, may experience particular challenges during the menopause transition as a result of having to manage menopausal symptoms in the context of other symptoms related to their long-term condition. However there are few data concerning the impact of the menopause in these populations.[10]

Existing data on menopause in WLHIV are scarce and often contradictory. There is no clear consensus on the impact of HIV status on age at menopause[11]; however, there is evidence that suggests that WLHIV experience a greater burden of menopausal symptoms than HIV-negative women, including vasomotor symptoms, sexual dysfunction and mood changes.[11] Furthermore, it is clear that WLHIV are at increased risk (compared with both HIV-negative women and HIV-positive men) of developing comorbid conditions such as osteoporosis and cardiovascular disease as a result of the synergistic effects of HIV and oestrogen depletion.[12 13]

However, there remain limited data on the menopause in WLHIV.[14] Much of the available data come from the USA and South America. Findings from these studies may not be applicable in settings such as sub-Saharan Africa and Europe, where patient cohorts differ in terms of ethnicity, comorbid conditions, substance misuse, socioeconomic status and healthcare access. There are limited or no data on the following in WLHIV: the impact of the menopause transition on QoL, sexual function, mental health, adherence to ART and retention in HIV care; the use of hormone replacement therapy (HRT) and other treatments; and the potential role of psychosocial interventions. Furthermore, there has been no qualitative research to date focusing on the lived experiences of reproductive ageing among WLHIV.

In response to these identified gaps in evidence, the PRIME Study (Positive Transitions Through the Menopause; www.ucl.ac.uk/prime-study) was designed to explore, for the first time in the UK, the impact of the menopause transition on the health and well-being of WLHIV.

The following are the specific research questions:

1. What is the prevalence of menopause (stratified by age) and menopausal symptoms among WLHIV?

2. What factors are associated with earlier age at menopause and increased menopausal symptoms in WLHIV?

3. Among WLHIV, what is the association between both menopausal status and symptoms, and (a) mental health, (b) sexual function, (c) QoL, (d) adherence to ART and (e) retention in HIV care?

4. What are the lived experiences of the menopause transition among WLHIV?

5. What is the current management of menopausal symptoms among WLHIV in the UK?

## METHODS AND ANALYSIS
### Overall study design
The PRIME Study is an ongoing, longitudinal, mixed-methods observational study of the impact of the menopause transition on the health and well-being of WLHIV. Mixed-methods research 'focuses on collecting, analysing, and mixing both quantitative and qualitative data in a single study or a series of studies'.[15] The underlying assumption of mixed-methods research is that it has the potential to address some research questions more comprehensively than by using either quantitative or qualitative methods alone.[15] We acknowledge the different epistemologies associated with quantitative and qualitative research, but draw on the philosophical tenets of pragmatism when seeking to combine these approaches.[16]

Nearly two-thirds of WLHIV in the UK are of black African ethnicity.[2] Understanding the impact of the menopause on health and well-being in an ethnically diverse population of WLHIV involves exploring clinical outcomes, individual beliefs about the menopause, the cultural construction of the postreproductive female body and lived experiences. The complex and multidimensional nature of the menopause disrupts a simple dichotomy of quantitative and qualitative methodologies, instead requiring an integration of both. The PRIME Study therefore draws on public health approaches to population health, and also medical anthropology, in particular the anthropology of gender and reproduction, in order to address our overall research question.

This work is theoretically informed by the concept of *intersectionality*, an analytic framework that seeks to understand how multiple social categories (in this case gender, ethnicity, age, menopausal status and HIV status) combine and intersect to shape experience and disadvantage.[17]

In this paper, we focus on the design of the baseline study and planned longitudinal follow-up. We have deployed a multiphase sequential mixed-methods study design (figure 1), combining exploratory and explanatory phases, with equal weight given to both quantitative and qualitative approaches. Baseline data collection has occurred in three phases (table 1), with some overlap due to time constraints.

Phase 1 was *qualitative*, comprising focus group discussions (FGDs) with WLHIV aged 45 and over, primarily to

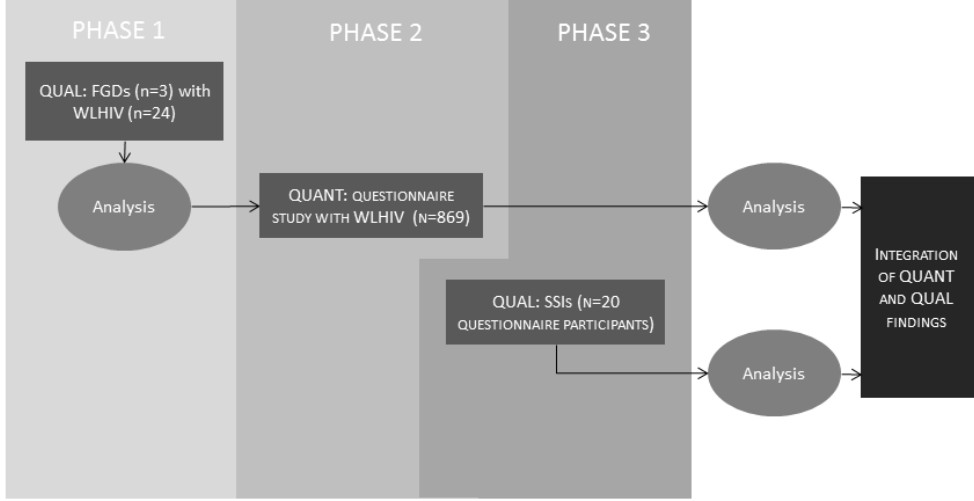

QUAL, qualitative; FGDs, focus group discussions; WLHIV, women living with HIV; QUANT, quantitative

**Figure 1** Overview of the PRIME Study design. PRIME, Positive Transitions Through the Menopause; SSIs, semistructured interviews.

inform the design of the quantitative questionnaire and semistructured interview (SSI) schedule (phases 2 and 3, respectively).

Phase 2 was *quantitative*, comprising a questionnaire study of WLHIV aged between 45 and 60 years attending for HIV care in England, collecting data on menopausal status, menopausal symptomatology and key clinical outcomes quantitatively. This was complemented by linkage to clinical data records (with participant consent). Prior to administering the quantitative questionnaire (phase 2), we conducted cognitive interviews with three WLHIV aged between 45 and 60 who were recruited via word of mouth through our existing professional networks (and who had not participated in phase 1). This qualitative approach allowed us to evaluate questionnaire items in terms of comprehension and question

interpretation, information retrieval and response elicitation, refining questions where necessary.[18]

Phase 3 was *qualitative* and comprised semistructured *qualitative* interviews with a subsample of women who participated in the phase 2 questionnaire study. This sequential design allows quantitative findings to be contextualised, while providing deeper understanding of the lived experiences of the menopause transition in WLHIV.

## Setting

In phase 1, we recruited WLHIV through Positively UK, the UK's leading HIV peer-support charity which is based in London, to participate in FGDs.

For the questionnaire and SSI study (phases 2 and 3), we recruited women attending one of the 21 National

**Table 1** Study phases and their relation to research questions among women living with HIV

| Research question | Phase | Sample | Data |
|---|---|---|---|
| 1. Prevalence of menopausal status and symptoms. | 1 | Questionnaire participants (clinic), n=869 | Questionnaire |
| 2. Factors associated with age at menopause and symptoms. | 1 | Questionnaire participants (clinic), n=869 | Questionnaire Clinical data |
| 3. Association between both menopausal status and symptoms, and mental health, sexual function, QoL, adherence to ART and retention in HIV care. | 1–3 | Community participants (HIV charity), n=24 (3 FGDs) Questionnaire participants (clinic), n=869 SSI participants (clinic), n=20 | FGD Questionnaire SSI |
| 4. Lived experiences of the menopause. | 1 and 3 | Community participants (HIV charity), n=24 (3 FGDs) SSI participants (clinic), n=20 | FGD SSI |
| 5. Current management of menopausal symptoms. | 1–3 | Community participants (HIV charity), n=24 (3 FGDs) Questionnaire participants (clinic), n=869 SSI participants (clinic), n=20 | FGD Questionnaire SSI |

ART, antiretroviral therapy; FGD, focus group discussion; QoL, quality of life; SSI, semistructured interview.

Health Service (NHS) clinics in England for HIV care. HIV care is available free of charge in the UK through specialist HIV clinics, and is where the overwhelming majority of PLHIV receive specialist care. We selected sites known to have large numbers of female patients in this age group (almost all of them in London) and sites that allowed for geographical diversity within England (G Rooney, Public Health England, personal communication, 13 February 2015).

### Phase 1: qualitative (FGDs)

The aim of phase 1 was to inform the design of the phase 2 questionnaire and phase 3 SSI schedule.

### Eligibility

In partnership with Positively UK, we invited WLHIV aged 45 and over (regardless of menopausal status) to participate in FGDs (purposive sampling). Women who did not speak English were unable to participate in FGDs. Working with two peer researchers (both WLHIV aged over 45), we approached Positively UK service users through advertising at Positively UK (by poster and flyers) and direct phone or face-to-face contact.

### Data collection

The first author (ST) conducted all FGDs on charity premises between June and August 2015, with peer researchers cofacilitating. Focus groups can be particularly useful in identifying group norms, exploring areas of consensus and dissent, and discussing potentially sensitive subjects (such as the menopause), especially among marginalised groups.[19] Topics explored included knowledge and experiences of the menopause (using a pile-sorting exercise to ascertain commonly experienced symptoms); language used to describe menopause and menopausal symptoms; care-seeking and self-management; and the impact of the menopause on their experience of living with HIV (and vice versa) (table 2). We also asked participants to comment on selected sections of a draft questionnaire (comprising validated questions obtained through an initial scoping review). This allowed us to ascertain whether questions were readily understood, easy to complete and acceptable to participants (this was particularly the case with sensitive questions such as about sexual function). This resulted in important changes. For instance, FGD data revealed that the term menopause

**Table 2** Focus group discussion and semistructured interview schedule

| Focus group discussion questions | Semistructured interview questions |
|---|---|
| 1. Can you share what you understand about what happens to women's periods as they get older? | 1. What do you understand by the word menopause? |
| 2. What happens to women's health as their periods begin to stop? | 2. Could you share any experiences you have of the menopause either personally or from other people? |
| 3. What do you understand by the word menopause? | 3. If perimenopausal or postmenopausal:<br>► How has life changed since the menopause (if at all)?<br>► How have you managed through this phase of life?<br>► How prepared did you feel for this phase of life?<br>► How do you think your experience of the menopause might be different from a woman without HIV?<br>► What is it like managing HIV through this phase of life? |
| 4. Pile-sorting exercise: menopausal symptoms. | 4. If premenopausal:<br>► What do you expect to happen to you during the menopause?<br>► How do you think life might change during this phase of life (if at all)?<br>► How prepared do you feel for this phase of life?<br>► How do you think experiences of the menopause might be different from a woman without HIV?<br>► How are you managing with HIV at the moment? |
| 5. Could you share any experiences you have of the menopause either personally or things you have heard from other people? | 5. All women:<br>► What could be done to help women during the menopause?<br>► Do women living with HIV need specific help, and if so what?<br>► Where do you think women living with HIV would like to go for help? |
| 6. If you had physical or emotional symptoms around this time, what could you do about it? | |
| 7. Do you think the menopause and HIV affect each other? | |
| 8. As you know we are doing some research on women living with HIV as they go through the menopause (when their periods stop as they older). What kind of things do you think we should be looking at? | |

was either not familiar to participants or had a variety of meanings. This allowed us to clarify the term in the final questionnaire. We also asked participants to highlight areas we had overlooked, for example, use of herbal remedies and the importance of social support. Each FGD lasted between 90 and 120 min, and was audio-recorded and transcribed verbatim.

## Sample size
Three FGDs (comprising 24 women in total, comprising 6–10 participants in each group) were deemed feasible within our timeframe and sufficient to achieve our aim.

## Phase 2: quantitative (questionnaires with linkage to clinical data)
The aim of phase 2 was to explore age at menopause, and the association between menopausal status and symptomatology, and key outcomes (objectives 1–3, and 5).

We conducted cognitive interviews with three WLHIV aged between 45 and 60 recruited from Positively UK and via the UK-CAB (the UK's HIV treatment advocate network) to inform questionnaire development. These interviews were conducted face-to-face by ST, audio-recorded and transcribed verbatim. We then proceeded to administer the questionnaires.

## Eligibility
Between January 2016 and June 2017, local clinical care teams recruited WLHIV (defined as female sex at birth for the purposes of this study) aged between 45 and 60 attending for HIV care at one of the 21 participating sites. Women were eligible to participate regardless of menopausal status. Women were *ineligible* if they had experienced surgical menopause or had a history of anything that might disrupt their bleeding pattern, such as hysterectomy; congenital absence of uterus and/or both ovaries; pregnancy or breast feeding within the last 12 months, and hormonal contraception within the last 6 months for either contraceptive or non-contraceptive use (women commencing intrauterine system as part of HRT *were* eligible); and chemotherapy or radiotherapy within the last 6 months. We also excluded women whose last menstrual period was more than 60 months prior (unless currently on HRT) as we aimed to capture women who were most likely to be experiencing symptoms. Women were not eligible if they were unable to give informed consent.

## Data collection
We administered self-completed paper questionnaires in English to all participants, taking approximately 15–30 min to complete. Participants were encouraged to complete the questionnaire within the clinic (with a private room offered to all participants); a minority chose to complete the questionnaire at home and return it by post. In cases of poor literacy or non-English speakers, participants were offered the opportunity to complete the questionnaire via a face-to-face interview, with the assistance of an interpreter (if required). The questionnaire

comprised 51 questions (subject to skip logic) divided into 5 sections: participant demographics; medical history and lifestyle factors; HIV-related history; menopausal status, symptoms and care-seeking; and sexual function. Where possible we used validated tools including the Menopause Rating Scale (MRS),[20] the Patient Health Questionnaire 4,[21] EuroQoL 5D (EQ-5D),[22] the National Survey of Sexual Attitudes and Lifestyle Sexual Function measure,[23] and the Hot Flash-Related Daily Interference Scale.[20] Menopausal status was determined from self-reported menstrual pattern (without biological confirmation), an approach that has been validated.[24] We categorised menopausal status according to the modified Stages of Reproductive Aging Workshop (STRAW) +10 criteria, internationally accepted staging criteria for menopausal status designed to facilitate comparability across studies.[25] With participants' consent, questionnaire data were supplemented by routinely collected clinical data, including nadir and current CD4 count, baseline and current HIV viral load, and current ART regimen.

## Sample size
The recruitment target was 1500 women (500 premenopausal, 500 perimenopausal and 500 postmenopausal). For primary outcomes, the key exposure variables are premenopausal status versus menopausal status (including both perimenopausal and postmenopausal women, where you would expect the greatest prevalence of menopausal symptoms). A sample size of 1500 provides at least 95% power at a 5% level of significance to compare key outcomes of interest, based on a ±10% difference from prevalence estimates from previous studies. For example adherence to ART among PLHIV in the UK is estimated to be 80%.[26] If we assume a rate of adherence to ART in the premenopausal group of 80% and 70% in the menopausal group, power will be >95%. Looking at a different outcome, if we assume a prevalence of depression of 25% in the premenopausal group[27] and 35% in the menopausal group, we will have 80% power to detect a difference.

## Phase 3: qualitative (SSIs)
### Eligibility
Women who completed the questionnaire in phase 2 and who gave consent to be contacted about a qualitative interview were contacted by telephone by ST and invited to participate in an SSI. Due to feasibility, women were only recruited from London sites. Sampling was purposive in that we recruited women of premenopausal, perimenopausal and postmenopausal status in order to reflect different stages of the menopause transition.

### Data collection
All SSIs were conducted by ST face-to-face in a private room on hospital or university premises between April 2016 and April 2018. ST is a female doctor and public health academic in her early 40s and is of British Pakistani ethnicity. Topics explored during SSIs include knowledge,

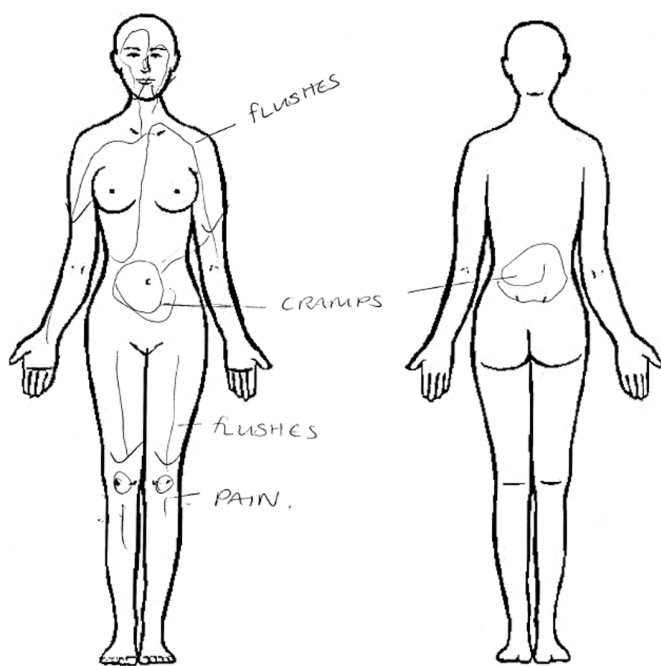

**Figure 2** Example of graphic elicitation of menopausal symptoms from a participant.

expectations and experiences of the menopause; cultural attitudes towards the menstruation and menopause; management of menopausal symptoms; impact of the menopause on health and well-being (including HIV care); and recommendations for interventions and service improvement (table 2). We also used *graphic elicitation*, asking women to represent their symptoms on a diagram of the body (figure 2). Graphic elicitation, the use of diagrams in interviews as stimuli, can yield data that may otherwise be difficult to obtain through verbal exchange, such as the experience of symptoms.[28] All SSIs were audio-recorded and transcribed verbatim.

### Sample size
A total of 20 women participated in SSIs, at which point data saturation was reached.

### Data analysis
#### Quantitative data analysis
Estimates of menopause prevalence stratified by age will be obtained using Kaplan-Meier plots, in which the cumulative proportion of participants who are postmenopausal will be plotted by age. Cox regression will be used to explore factors associated with age at menopause, such as nadir CD4 count. For analyses of associations between key outcomes and (1) menopausal status and (2) menopausal symptoms, we will use $\chi^2$ tests and t-tests (or a non-parametric equivalent) to compare characteristics in each group, followed by multivariable logistic regression modelling. We define menopausal status according to the modified STRAW+10 criteria[25] as follows:

premenopausal (menses within the past 3 months, no interval of amenorrhoea for ≥60 days in the past 6 months and no late period in the past 2 years); perimenopausal (menses within the past 12 months but an interval of amenorrhoea for ≥60 days in the past 6 months and/or a late period in the past 2 years); and postmenopausal (amenorrhoea for ≥12 months). Menopausal symptoms will be captured using the MRS and categorised according to standard cut-offs.[20]

#### Qualitative data analysis
We will analyse FGD and SSI data thematically, using an integrated approach that includes both inductive development of codes as well as a deductive organising framework (derived from results from the quantitative analyses).[29] This will facilitate integration of quantitative and qualitative data. For instance, if low sexual function is noted to be prevalent among women of postmenopausal status, then we will interrogate our qualitative data sets to explore attitudes towards or experiences of reduced sexual function among participants. Transcripts will be read several times with sections coded. Coded text will then be compared and linked across other interviews if they capture similar themes, leading to the development of broader key categories.

#### Mixed-methods data integration
This study draws on quantitative data from questionnaires and qualitative data from FGDs and SSIs. In order to maximise the potential of this rich mixed-methods data set, we will integrate findings from each data source, following analyses of quantitative and qualitative data. This will allow us to contextualise findings, identify areas of discrepancy and generate new hypotheses. Quantitative and qualitative data will be kept analytically distinct and analysed using the approaches outlined above. We will primarily integrate findings using *convergence code matrices* (table 3).[30] This approach allows us to triangulate findings from each phase of the study in relation to each specific research objective, highlighting areas of agreement, disagreement or silence (in cases where one data set makes no reference to this objective). Furthermore, as we will have both quantitative and qualitative data on a subset of participants (those who participated in an SSI), we will be able to construct mixed-methods cases,[31] using matrices to visualise quantitative and qualitative data from each participant on menopausal symptoms, mental health, sexual function, QoL, adherence to ART and retention in HIV care (table 4).

This again will allow us to explore similarities and discrepancies in data from each participant, as well as to identify patterns *across* individuals. Understanding areas of complementarity and divergence in our mixed-methods data set is critical in gaining further insight into findings.

**Table 3** Example of convergence coding matrix for quantitative and qualitative data collection

| Impact of menopausal symptoms on | QUANT findings | QUAL findings | Integrated findings (agreement/partial agreement/ silence/disagreement) |
|---|---|---|---|
| Mental health | | | |
| Sexual function | | | |
| Quality of life | | | |
| Adherence to antiretroviral therapy | | | |
| Retention in HIV care | | | |

QUAL, qualitative; QUANT, quantitative.

## Longitudinal follow-up

In October 2018, we will undertake a follow-up study of 100 PRIME participants. We aim to describe levels of follicle stimulating hormone (FSH) in WLHIV aged >45 who were categorised as perimenopausal at baseline and who now report amenorrhoea for ≥12 months (and therefore would be categorised at postmenopausal). FSH levels in the general population increase significantly from premenopause to postmenopause.[32] However, national guidelines advise that in women aged >45 years, menopause is a clinical diagnosis, established after 12 or more months of amenorrhoea in those with an intact uterus and not using hormonal contraception.[6] Disordered menstrual function and prolonged amenorrhoea (in the absence of biological markers of ovarian failure) have been reported in HIV, but in older studies comprising women who were diagnosed with advanced HIV and had limited access to ART.[33] Data on FSH levels from our longitudinal follow-up will allow us to confirm if menstrual history in WLHIV aged 45–60 is sufficient for diagnosing menopause in the contemporary ART era.

We will recruit 100 PRIME participants who were categorised as perimenopausal at entry into the cohort (between January 2016 and June 2017), who provided consent for ongoing contact about PRIME-related studies and who now report ≥12 months amenorrhoea. We will administer a short self-completed paper questionnaire (to ascertain menopausal symptoms and mental health), collect data on HIV viral load from clinical databases and take a serum sample to measure FSH.

A previous study of FSH in postmenopausal women (without HIV) demonstrated that the 25th centile for FSH in this group was 30 mIU/mL.[32] Cejtin et al,[34] in their study of FSH levels in WLHIV aged 16–55 with amenorrhoea, revealed a prevalence of elevated FSH (defined as >25 mIU/mL) of 47%. We therefore believe 60% to be a pragmatic and conservative estimate of prevalence of elevated FSH in this age group. Based on this estimated prevalence, a sample size of 96 will allow us to obtain an estimate with 10% precision.

### Patient and public involvement

We are committed to the meaningful involvement of WLHIV at *all* stages of the research process[35] and have drawn on National Institute for Health Research (NIHR) INVOLVE guidance.[36] The study proposal was discussed and reviewed by WLHIV. We recruited three community representatives (all WLHIV aged 45 or over) to sit on our Expert Advisory Group via the UK-CAB, alongside academics and clinicians. The community representatives provide insight and expertise into the design, conduct, dissemination and development of the study. Two of the community representatives also worked as peer researchers in phase 1 of the study, providing invaluable expertise in recruitment, facilitation of focus groups and analysis of data.

## ETHICS AND DISSEMINATION

Written informed consent has been obtained from participants in all phases of the baseline study. We have also sought consent from baseline questionnaire participants to contact them in the future about other studies related to the PRIME Study. SSI and FGD (but not questionnaire) participants were reimbursed with a £20 shopping voucher in recognition of their time.

**Table 4** Example of convergence coding matrix for analysing quantitative and qualitative data within and across individual participants

| Participant ID | QUANT: psychological distress (PHQ-4 >3) | QUAL: impact of menopausal symptoms on mental health | QUANT: menopause care-seeking | QUAL: menopause care-seeking |
|---|---|---|---|---|
| 1 | | | | |
| 2 | | | | |
| 3 | | | | |

PHQ-4, Patient Health Questionnaire 4; QUAL, qualitative; QUANT, quantitative.

Quantitative and qualitative data collected during the study are not open access. However, they can be made available by request to the study team. In conjunction with the Expert Advisory Group, we have established a clear dissemination strategy that takes into account a wide range of stakeholders including WLHIV. A publication plan includes presentation at scientific conferences and peer-reviewed publications (at least five empirical papers). Where possible, we will present quantitative and qualitative data together, drawing on the Good Reporting of a Mixed Methods Study framework.[37] We have synthesised initial study findings in an accessible report aimed primarily at a non-technical audience. This report was launched at a live-streamed dissemination event aimed at key stakeholders including WLHIV. Finally, throughout the study, we have been highlighting progress and emergent findings through our study website (www.ucl.ac.uk/prime-study) and study Twitter account (@PRIME_UCL).

## DISCUSSION

This ongoing mixed-methods observational study comprising FGDs, questionnaires and SSIs aims to examine the impact of menopause on the health and well-being of WLHIV. It was designed in response to a recognised paucity of data in this growing patient group.

### Limitations

Our original target recruitment number for phase 2 was 1500 WLHIV, based on an ineligibility rate of 10%. Despite an uptake of 80% among women approached, we were not able to reach this target due to a much higher ineligibility rate than anticipated. In total, 1999 women were approached, of whom 1312 (65.6%) were eligible to participate. Of these eligible women, 1059 (80.7%) consented to participate; we have completed questionnaires on 869 women. However, a sample size of 869 still provides at least 80% power at a 5% level of significance to compare key outcomes of interest, based on a ±10% difference from prevalence estimates.

We are aware of the potential for selection bias; however, it is hard to predict whether women with increased menopausal symptoms would be more or less likely to participate in the study. In recruiting solely from NHS sites in phases 1 and 2, we will have excluded the minority of women who are not engaged in HIV care, whose needs and experiences may differ. In recruiting women aged between 45 and 60, we will not have data on those who have experienced premature ovarian insufficiency (menopause aged <40 years). One of our exclusion criteria is use of hormonal contraception, which may have led to sexually active women being more likely to be excluded. Of those who were ineligible and on whom we have data on ineligibility, 23% were on hormonal contraception. The most common reason for ineligibility was last menstrual period more than 60 months prior.

In terms of study design, the cross-sectional nature of the baseline PRIME Study means we are unable to infer causality in analyses of these quantitative data. Finally, the lack of an HIV-negative comparison group prevents us from looking at the impact of HIV status on the experience of menopause. However, in future work we plan to explore ways of recruiting a similarly aged and ethnically diverse group of HIV-negative women.

### Strengths

A key strength of the PRIME Study lies in its combination of quantitative and qualitative data on reproductive ageing in WLHIV. This mixed-methods approach will allow us to address our research questions more comprehensively than had we used either quantitative or qualitative methods alone. Having recruited 869 participants in phase 2, with ongoing qualitative data collection in a subset of these women, the PRIME Study is already the largest study of menopause in WLHIV in Europe and one of the largest globally. We have recruited women from clinics *across* England, which means study findings are likely to be generalisable nationally. The questionnaire was designed in discussion with ongoing women's cohort studies in North America and with the Third National Survey of Sexual Attitudes and Lifestyles, a national probability sample survey of sexual behaviour and attitudes in England. This has allowed us to incorporate validated tools used in these other studies, enabling future combined analyses. Finally, we will be following a subset of PRIME participants longitudinally from October 2018, with further longitudinal follow-up planned in the future subject to funding.

## CONCLUSION

Increasing availability and effectiveness of ART means that a growing number of WLHIV will be reaching mid-life and beyond, both in the UK and globally. The needs of WLHIV transitioning through the menopause are currently poorly understood. The PRIME Study is therefore timely in its focus on the menopause in this patient population. We anticipate that our study findings will produce new insights into how the menopause impacts their health and well-being, informing the commissioning and provision of services for WLHIV *across* the life course.

**Acknowledgements** We would like to acknowledge the PRIME Study Expert Advisory Group: Comfort Adams, Jane Anderson, Mwenza Blell, Jonathan Elford (Chair), Janine MacGregor-Read, Fiona Pettitt, Janice Rymer, Jane Shepherd, Lorraine Sherr and Emily Wandolo; and the PRIME Study recruiting sites: Barking Community Hospital (Rageshri Dhairyawan, Emma Macfarlane, Sharmin Obeyesekera, Cecelia Theodore); Barts Hospital NHS Trust (James Hand, Helena Miras, Liat Sarner); Brighton and Sussex University Hospital (Yvonne Gilleece, Alyson Knott, Celia Richardson); Chelsea and Westminster Hospital (Mimi Chirwa, Ann Sullivan, Mini Thankachen, Sathya Visvendra); City of Coventry Health Centre (Sris Allen, Kerry Flahive); Guy's and St Thomas' Hospital (Julie Fox, Julianne Iwanga, Annemiek DeRuiter, Mark Taylor); 10 Hammersmith Broadway (Sophie Hobday, Rachael Jones, Clare Turvey); Homerton University Hospital (Monica James, Sambasivarao Pelluri, Iain Reeves); Kings College Hospital (Sarah Barber, Priya Bhagwandin, Lucy Campbell, Leigh McQueen, Frank Post, Selin Yurdakul, Beverley White); Lewisham and Greenwich NHS Trust (Tarik Moussaoui, Melanie Rosenvinge, Judith Russell); Mortimer Market Centre (Tuhina Bhattacharya, Alexandra Rolland, Shema Tariq); New Cross Hospital Wolverhampton (Sarah Milgate, Anjum Tariq); North Manchester General Hospital (Claire Fox, Gabriella Lindergard, Andrew

Ustianowski); Royal Free Hospital NHS Trust (Fiona Burns, Nargis Hemat, Nnenna Ngwu, Rimi Shah); Southend Hospital (Sabri Abubakar, John Day, Laura Hilton, Henna Jaleel, Tina Penn); St Mary's Hospital (Angela Bailey, Nicola Mackie); University Hospital Birmingham (Reka Drotosne-Szatmari, Jan Harding, Satwant Kaur, Tessa Lawrence, Monika Oriak, Jonathan Ross); and West Middlesex Hospital (Kimberley Forbes, Ursula Kirwan, Shamela De Silva, Marie-Louise Svensson, Rebecca Wilkins). We are grateful for support from Positively UK and the UK-CAB. Finally, and most importantly, we thank all our participants for sharing their time and experiences so generously with us.

**Contributors** ST conceived and designed the study with support from FMB, RG and CS. ST drafted the first version of this article. All authors critically reviewed the first version of the article and approved the final draft for publication.

**Funding** "The baseline PRIME study was funded in the form of a National Institute of Health Research (NIHR) postdoctoral fellowship awarded to ST, funded by the National Institute of Health Research (PDF-2014-07-071). The follow-up study is funded by the British HIV Association (BHIVA) in the form of a BHIVA Research Award commencing in September 2018. Between March and December 2018 ST received salary support to continue work on this study, through a UCL/Wellcome Institutional Strategic Support Fund Flexible Support Award (204841/Z/16/Z). This manuscript presents independent research funded by the NIHR."

**Disclaimer** The views expressed are those of the authors and not necessarily those of the NHS, the NIHR or the Department of Health.

**Competing interests** ST has previously received a travel bursary funded by Janssen-Cilag through the British HIV Association, and speaker honoraria and funding for preparation of educational materials from Gilead Sciences. ST, FMB and CS are members of the steering group of SWIFT, a networking group for people involved in research in HIV and women, funded by Bristol-Myers Squibb. CS has received funding for membership in Data Safety and Monitoring Boards, Advisory Boards, speaker panels and for preparation of educational materials from Gilead Sciences, ViiV Healthcare and Janssen-Cilag. FMB has received consultancy fees and conference support from Gilead Sciences.

**Patient consent for publication** Not required.

**Ethics approval** Qualitative research undertaken outside the NHS in phase 1 was reviewed by the University College London Research Ethics Committee (Project ID: 6698/001). The baseline study has ethical approval from the South East Coast-Surrey Research Ethics Committee on behalf of all NHS sites (REF 15/0735) for phases 2 and 3, and approval from the South Central Hampshire Research Ethics Committee (REF 18/SC/0570) for the follow up study.

**Provenance and peer review** Not commissioned; externally peer reviewed.

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
