## [Reviewer comments · BMJ Open]

ARTICLE DETAILS

TITLE (PROVISIONAL)	The PRIME (Positive Transitions Through the Menopause) Study: a protocol for a mixed-methods study investigating the impact of the menopause on the health and well-being of women living with HIV in England
AUTHORS	Tariq, Shema; Burns, Fiona M.; Gilson, Richard; Sabin, Caroline

VERSION 1 - REVIEW

REVIEWER	Adebola Adedimeji Albert Einstein College of Medicine, Bronx, NY, USA
REVIEW RETURNED	13-Aug-2018

GENERAL COMMENTS	Authors need to clarify statements in the methods. For instance, on page 6, lines 31-43, it is not clear if the participants selected for the cognitive interviewing were part of phase 1. Authors also need to justify the rationale for using cognitive interviewing. More importantly, authors should clarify if the results of the cognitive interviewing informed the development of the questionnaire It is also unclear if age was the only criterion for eligibility as described on page 7, lines 21-24 Authors need to clarify the sentence described on page 10, lines 25-31 regarding the development of a priori codes. What is the modality for doing this?
--

REVIEWER	Anna Rubtsova Emory University, USA
REVIEW RETURNED	20-Sep-2018

GENERAL COMMENTS	This is a much needed study of menopause and its consequences among women living with HIV (WLHIV). The mixed-methods methodology is a strength given the cultural underpinnings of menopause experience. I think that the protocol of this exciting study will make a contribution to the literature. My comments mostly ask for clarification of relevant details. Introduction The introduction would benefit from a more thorough description of the current state of research related to the five specific research questions examined by the PRIME study. For example,
---

introduction provides no information about what is already known or the gaps in research related to the factors associated with earlier age of menopause and increased menopausal symptoms among WLHIV, even though research question # 2 is designed to explore these factors. Similarly, the introduction would benefit from adding more information about the current management of menopausal symptoms among WLHIV in the UK investigated by the research question # 5.

Overall study design

- This study has a complex design, with different phases using different samples and methodology. It is also not quite clear how these different phases and methods relate to the research questions. To help guide readers through this complexity, consider adding a table, which might contain the following columns:

Research Question, Research Phase, Sample, Data Source, Analysis.

- Currently, the manuscript provides no information on recruitment methods (e.g., direct recruitment, flyers, referrals?), participant compensation (except for FGD and SSI), or sampling methods (except for SSI). It would be helpful to provide these details for each of the three phases and the longitudinal follow-up.

Phase 1: Qualitative (focus group discussions)

- Were there any exclusion criteria for the FGDs? For example, were participants excluded based on the limited English language proficiency?

- It would be helpful to provide examples of FGD questions. Consider adding a table with a topic guide, where you could list FGD topics in one column and examples of related questions in the other. (It would also be helpful to provide a similar topic guide with question examples for SSIs in this or a separate table).

- Please, specify the number of participants per a focus group.

Phase 2: Quantitative (questionnaires with linkage to clinical data)

- The description of paper questionnaire would benefit from providing the following details: the number of questions, approximate time to complete, and the setting where these questionnaires were administered (e.g., were they completed by women during their clinical visit? Were they provided a private space in the clinic to complete it?)

- The manuscript states that the aim of the Phase 1 was to inform the design of the Phase 2 questionnaire and Phase 3 SSI schedule. However, from the current description of the paper questionnaire, I am under the impression that it mostly consisted of the standardized validated scales. It is not clear how exactly the results of the focus groups informed the design of paper questionnaire. Did the FGDs inform the selection of particular scales? Did you design certain questions yourself based on the results of the FGDs? I think it would be helpful to provide a more explicit discussion of the links between the results of Phase 1 and the design of Phase 2.

- Since the age of menopause is one of the key variables in Phase 2 and since it is known to be difficult to determine in research, consider adding several sentences to the following description on page 8, line 47 "Menopausal status was determined from self-reported menstrual pattern." Were these questions used by other published research? If not, it would be helpful to provide a brief description of the questions used to determine menstrual pattern.

REVIEWER	Prof Abigail Locke University of Bradford, UK
REVIEW RETURNED	19-Oct-2018

GENERAL COMMENTS	Thank you for submitting this protocol for review. The proposal concerns a mixed-methods study – PRIME (Positive Transitions Through the Menopause) – that concerns menopause in women living with HIV in England. As the authors note, before they began their work, there was no published record of qualitative investigation of menopausal women with HIV. The protocol locates the study and the need for this work within existing literature. The area of study is an interesting and important one. My comments on this piece are on the specific research protocol that I have been asked to consider. The protocol is split into three main phases of data collection. It appears that all three phases have been completed, but the dates are not apparent for phase three. However, there is a follow up quantitative phase of 100 women from October 2018 to September 2019 that does not form part of these 3 phases. I will discuss aspects of each phase of the study but first I have some general comments onto the protocol. They argue that a mixed-methods approach is more suitable to this area of study and stronger than either quantitative or qualitative approaches alone. I would dispute this and would suggest that all of the approaches have something to offer the topic area. Combining the approaches into a mixed methods take carries with it some benefits but also some potential for diluting the depth of the qualitative analysis, if, for example, the findings are not fully explicated. I was pleased to see the inclusion by the authors of the need to include intersectionality as a lens by which to consider the data. It would be useful to see how that information is being made explicit in the analysis of data. I would like to ask them on the decision to exclude women who were not receiving care. The justification given for this in the protocol was that their needs would differ from the sampled group. I am sure that this would be the case. However, the experiences and needs of these women need to be gathered. Therefore, I would suggest that a secondary piece of work picked up this particular issue. The question would be – why are these women not accessing services or support? Whilst the study is a mixed-methods design, the theoretical framework for the protocol was not made explicit. Theoretical frameworks set the tone for research studies and work out the types of questions that can be asked of the data sets and the way in which the analyses are interpreted. The three phases appear to be using potentially different theoretical frameworks. It would be useful, moving forward, for the authors to make these frameworks accessible. For example, it appears that phases 1 and 2 are sitting within a realist framework based on an objective epistemology, as does much quantitative research. However, with the mention of 'lived experiences' as the novel basis for the qualitative aspects of the study, it appears that a more interpretative lens is being applied, implying a more subjectivist epistemology. In Phase 3, the authors claim to be using the qualitative methodology of Grounded Theory, this is a methodology that can be used in a variety of ways and with a variety of theoretical backgrounds. However, it would have been useful for the authors to locate it within a particular theoretical perspective. As this is a protocol, and not an actual
--

	empirical piece with findings, it is not possible to ascertain how successful this data will be at accessing lived experiences. Moving through each of the phases of the protocol, the authors note that phase 1 was carried out to set up part of phase 2 and 3. In phase 1, the authors carried out three focus groups and discussed a variety of topics. They note that these were qualitatively analysed but do not provide information on the type of qualitative analysis, and the theoretical framework on which it rested. This moves into phase 2, which is the quantitative questionnaire based part. The authors claim that qualitative data fed into the creation of the questionnaire in phase 2 with the addition of cognitive interviews with three participants that fed into the process. What is not altogether clear from the protocol is how phase 1 with the focus groups fed into phase 2. Particularly as the authors note that draft questionnaire questions were fed into the focus groups. Therefore, it appears that these two phases were feeding into each other, rather than being discrete phases, although this does not work with the timescales for each of these that were given? With that in mind, it would be useful to know where the draft questionnaire information originated from if it fed into phase 1. Secondly, I would be interested into the inclusion of the validated pre-existing measures in the questionnaire data of phase 2, and again, how this links to the qualitative input from phase 1. Phase 3, as noted previously, is an interview study of 20 participants and analysed using a version of Grounded Theory. I was interested in their claim that some a priori codes were added into the analysis. Typically qualitative analysis is inductive, there are a few methodologies that code using a priori codes, but I am interested to explore the combination of both a little more. It is not altogether clear whether phase 3 has been completed, or the data collection has been completed, whilst the analysis has not. No dates are given for this aspect of work but we can infer given that the authors use past tense for data collection but future tense for the qualitative analysis. Moreover, we are informed that there is a follow-up phase that is taking place from October 2018 – September 2019 encompassing follow up with 100 women. The authors note some aspects of consideration in the follow up but the actual methods of collecting and analysing that data would benefit from additional clarity. Again, without a more detailed commentary on the theoretical underpinnings behind the analysis, given that almost the entire protocol has now been conducted, it is difficult to comment further, although I do see the need for the study overall and the benefits of conducting it with this particular group of women.
--	--

VERSION 1 – AUTHOR RESPONSE

Reviewer 1

1. On page 6, lines 31-43, it is not clear if the participants selected for the cognitive interviewing were part of phase 1.

This is now clarified.

2. Authors also need to justify the rationale for using cognitive interviewing. More importantly, authors should clarify if the results of the cognitive interviewing informed the development of the questionnaire

This is already justified on page 6 as follows “This qualitative approach allowed us to evaluate questionnaire items in terms of comprehension and question interpretation, information retrieval, and response elicitation, refining questions where necessary (17)”.

3. It is also unclear if age was the only criterion for eligibility as described on page 7, lines 21-24.

We state eligibility as follows “WLHIV aged 45 and over (regardless of menopausal status)”. We have now added that women unable to speak English were excluded from FGDs.

4. Authors need to clarify the sentence described on page 10, lines 25-31 regarding the development of a priori codes. What is the modality for doing this?

Now clarified by citing a hypothetical example and citing a reference.

Reviewer 2

1. The introduction would benefit from a more thorough description of the current state of research related to the five specific research questions examined by the PRIME study

We feel a thorough review of the literature around menopause and HIV is beyond the scope of a study protocol. However we do summarise gaps in data (specifically addressing our research questions) and reference our review paper (reference 11) which synthesises current data in detail.

2. This study has a complex design, with different phases using different samples and methodology. It is also not quite clear how these different phases and methods relate to the research questions. To help guide readers through this complexity, consider adding a table, which might contain the following columns: Research Question, Research Phase, Sample, Data Source, Analysis.

We agree and now include this.

3. Currently, the manuscript provides no information on recruitment methods (e.g., direct recruitment, flyers, referrals?), participant compensation (except for FGD and SSI), or sampling methods (except for SSI). It would be helpful to provide these details for each of the three phases and the longitudinal follow-up.

This is now clarified in the manuscript.

4. Were there any exclusion criteria for the FGDs? For example, were participants excluded based on the limited English language proficiency?

This is now clarified in the manuscript.

5. It would be helpful to provide examples of FGD questions. Consider adding a table with a topic guide, where you could list FGD topics in one column and examples of related questions in the other. (It would also be helpful to provide a similar topic guide with question examples for SSIs in this or a separate table).

We are unable to provide this due to limited word count. We have included topics covered. We would have included both the FGD and SSI topic guides as supplementary material but cannot see that this an option for study protocols.

6. Please, specify the number of participants per a focus group.

Now included.

7. The description of paper questionnaire would benefit from providing the following details: the number of questions, approximate time to complete, and the setting where these questionnaires were administered (e.g., were they completed by women during their clinical visit? Were they provided a private space in the clinic to complete it?)

These details have been added.

8. The manuscript states that the aim of the Phase 1 was to inform the design of the Phase 2 questionnaire and Phase 3 SSI schedule. However, from the current description of the paper questionnaire, I am under the impression that it mostly consisted of the standardized validated scales. It is not clear how exactly the results of the focus groups informed the design of paper questionnaire. Did the FGDs inform the selection of particular scales? Did you design certain questions yourself based on the results of the FGDs? I think it would be helpful to provide a more explicit discussion of the links between the results of Phase 1 and the design of Phase 2.

Now provided with clear examples of changes made as a result of Phase 1.

9. Since the age of menopause is one of the key variables in Phase 2 and since it is known to be difficult to determine in research, consider adding several sentences to the following description on page 8, line 47 "Menopausal status was determined from self-reported menstrual pattern." Were these questions used by other published research? If not, it would be helpful to provide a brief description of the questions used to determine menstrual pattern.

We have now clarified that self-reported menstrual pattern has been validated to assess menopausal status, and have stated that we use STRAW+10 criteria.

Reviewer 3

1. The protocol is split into three main phases of data collection. It appears that all three phases have been completed, but the dates are not apparent for phase three.

Now included in manuscript.

2. They argue that a mixed-methods approach is more suitable to this area of study and stronger than either quantitative or qualitative approaches alone. I would dispute this and would suggest that all of the approaches have something to offer the topic area. Combining the approaches into a mixed methods take carries with it some benefits but also some potential for diluting the depth of the qualitative analysis, if, for example, the findings are not fully explicated.

We believe that there is strength in combining quantitative and qualitative approaches, as opposed to conducting either a quantitative or qualitative study alone. Analytic depth is not necessarily compromised by this approach. We refer to Reviewer 2 who states a mixed methods approach is a strength.

3. I was pleased to see the inclusion by the authors of the need to include intersectionality as a lens by which to consider the data. It would be useful to see how that information is being made explicit in the analysis of data.

We believe that this is only possible to demonstrate how intersectionality will be used in data analysis through the presentation of empirical data, and is hard to demonstrate in a study protocol.

4. I would like to ask them on the decision to exclude women who were not receiving care. The justification given for this in the protocol was that their needs would differ from the sampled group. I am sure that this would be the case. However, the experiences and needs of these women need to

be gathered. Therefore, I would suggest that a secondary piece of work picked up this particular issue. The question would be – why are these women not accessing services or support?

This is a study based within a clinic setting. Therefore by definition all women were accessing care. We do not use their differing needs as justification, rather we state this as a limitation of our work. We agree women who have disengaged from care are an important group, but this is a difficult group to access as they do not present to clinic or to support services. This is a limitation even in studies whose focus is retention in HIV care. We agree that this would be worthy of further study.

5. Whilst the study is a mixed-methods design, the theoretical framework for the protocol was not made explicit.

We state that the work has been informed by intersectionality theory. We hope that the concerns about the theoretical underpinnings to our mixed-methods approach are addressed below.

6. The three phases appear to be using potentially different theoretical frameworks. It would be useful, moving forward, for the authors to make these frameworks accessible. For example, it appears that phases 1 and 2 are sitting within a realist framework based on an objective epistemology, as does much quantitative research. However, with the mention of 'lived experiences' as the novel basis for the qualitative aspects of the study, it appears that a more interpretative lens is being applied, implying a more subjectivist epistemology.

The combining of epistemologies in mixed methods research has been well-debated in the mixed methods literature and we feel a detailed discussion of this is beyond the scope of this study protocol. We now include the following to highlight the differences in epistemologies and the philosophical assumptions underpinning our approach, citing a methodology paper: "We acknowledge the different epistemologies associated with quantitative and qualitative research, but draw upon the philosophical tenets of pragmatism when seeking to combine these approaches."

7. In Phase 3, the authors claim to be using the qualitative methodology of Grounded Theory, this is a methodology that can be used in a variety of ways and with a variety of theoretical backgrounds. However, it would have been useful for the authors to locate it within a particular theoretical perspective.

On reflection we realise that this work is most suited to thematic analysis and have amended accordingly.

8. As this is a protocol, and not an actual empirical piece with findings, it is not possible to ascertain how successful this data will be at accessing lived experiences.

We agree and look forward to presenting our empirical findings.

9. In phase 1, the authors carried out three focus groups and discussed a variety of topics. They note that these were qualitatively analysed but do not provide information on the type of qualitative analysis, and the theoretical framework on which it rested.

We have clarified that FGD and SSI data are analysed in similar ways (qualitative data analysis section).

10. The authors claim that qualitative data fed into the creation of the questionnaire in phase 2 with the addition of cognitive interviews with three participants that fed into the process. What is not altogether clear from the protocol is how phase 1 with the focus groups fed into phase 2.

Now clarified, see response to reviewer 2.

11. With that in mind, it would be useful to know where the draft questionnaire information originated from if it fed into phase 1.

Clarified that we had some validated measures obtained through an initial scoping review.

12. Secondly, I would be interested in the inclusion of the validated pre-existing measures in the questionnaire data of phase 2, and again, how this links to the qualitative input from phase 1.

Similar to point 10, this is now clarified including examples of how the questionnaire was adapted as a result of Phase 1.

13. Phase 3, as noted previously, is an interview study of 20 participants and analysed using a version of Grounded Theory. I was interested in their claim that some a priori codes were added into the analysis. Typically qualitative analysis is inductive, there are a few methodologies that code using a priori codes, but I am interested to explore the combination of both a little more.

We used this approach in order to facilitate dialogue between quantitative and qualitative datasets. We now explain this in more detail and include a hypothetical example as well as a reference.

14. It is not altogether clear whether phase 3 has been completed, or the data collection has been completed, whilst the analysis has not.

Now clarified.

15. Moreover, we are informed that there is a follow-up phase that is taking place from October 2018 – September 2019 encompassing follow up with 100 women. The authors note some aspects of consideration in the follow up but the actual methods of collecting and analysing that data would benefit from additional clarity

This is now summarised.

VERSION 2 – REVIEW

REVIEWER	Adebola Adedimeji Albert Einstein College of Medicine
REVIEW RETURNED	07-Feb-2019

GENERAL COMMENTS	Manuscript is suitable for acceptance having addressed all the reviewer's concerns
--

REVIEWER	Anna Rubtsova Emory University, USA
REVIEW RETURNED	08-Feb-2019

GENERAL COMMENTS	It was my pleasure reviewing this manuscript. I think authors did a nice job responding to reviewers' comments. I recommend publishing with the following minor change: please, include samples of FGD and SSI questions either as a table or within the manuscript text as part of respective sections on FGDs and SSIs. For a protocol of a qualitative or a mixed methods study, providing an interview guide is extremely important since it contributes to
---

	study reproducibility, and a word limit cannot be a deterrent. Slightly editing down the text to decrease the word count, or including sample questions as a Table or as an Appendix are all viable options.
--	--

REVIEWER	Abigail Locke Professor of Psychology, University of Bradford, UK.
REVIEW RETURNED	25-Feb-2019

GENERAL COMMENTS	This is a much improved revision. My previous review noted the lack of clarity in terms of description of study stages, qualitative methods of analysis and theoretical frameworks. The authors have addressed each of these points in turn. There is still more that could be said around theoretical perspectives and mixed-methods approaches. However, this may be for another paper. I look forward to hearing more about the study.
---

VERSION 2 – AUTHOR RESPONSE

Reviewer 2

1. We have added a table summarising both the focus group and interview schedule.